# Probing the Nanostructure of Neutron-Irradiated Diamond Using Raman Spectroscopy

**DOI:** 10.3390/nano10061166

**Published:** 2020-06-15

**Authors:** Andrey A. Khomich, Roman A. Khmelnitsky, Alexander V. Khomich

**Affiliations:** 1Kotelnikov Institute of Radio-Engineering and Electronics of the Russian Academy of Sciences, pl. Vvedenskogo 1, 141190 Fryazino, Russia; khmelnitskyra@lebedev.ru (R.A.K.); alex-khomich@mail.ru (A.V.K.); 2Lebedev Institute of Physics of the Russian Academy of Sciences, Leninsky pr. 53, 117924 Moscow, Russia

**Keywords:** diamond crystal, neutron irradiation, annealing, Raman spectra, phonon confinement, boson peak, defects

## Abstract

Disordering of crystal lattice induced by irradiation with fast neutrons and other high-energy particles is used for the deep modification of electrical and optical properties of diamonds via significant nanoscale restructuring and defects engineering. Raman spectroscopy was employed to investigate the nature of radiation damage below the critical graphitization level created when chemical vapor deposition and natural diamonds are irradiated by fast neutrons with fluencies from 1 × 10^18^ to 3 × 10^20^ cm^−2^ and annealed at the 100–1700 °C range. The significant changes in the diamond Raman spectra versus the neutron-irradiated conditions are associated with the formation of intrinsic irradiation-induced defects that do not completely destroy the crystalline feature but decrease the phonon coherence length as the neutron dose increases. It was shown that the Raman spectrum of radiation-damaged diamonds is determined by the phonon confinement effect and that the boson peak is present in the Raman spectra up to annealing at 800–1000 °C. Three groups of defect-induced bands (first group = 260, 495, and 730 cm^−1^; second group = 230, 500, 530, 685, and 760 cm^–1^; and third group = 335, 1390, 1415, and 1740 cm^−1^) were observed in Raman spectra of fast-neutron-irradiated diamonds.

## 1. Introduction

Radiation physics is one of intensively developing fields of research and application of diamonds; the interest in this subject is due to the following reasons. First, by most criteria, due to the record-high energy of interatomic bonds, the absence of the ionic component of bonds, and the negligible role of sub-threshold defect formation processes, diamond exhibits the highest resistance to various kinds of ionizing radiation, exceeding in this respect silicon by 1.5–2 orders of magnitude [1]. High radiation resistance becomes a crucial factor when a detector of ionizing radiation is exposed to intense radiation. Second, diamond is one of allotropes of carbon, which are known for their wide spectrum of physicochemical properties and find more and more new applications. Radiation damage (RD) of diamond followed by thermal annealing is an efficient method of defect engineering, which allows one to implement completely carbon structures with unique parameters. Under radiation damage above the graphitization threshold, diamond can transform into a stable graphite-like phase. This feature of diamond has been suggested to be used in micro- and acousto-electronics [2]. Third, owing to the high Debye temperature, room temperature is “low” for diamond, in contrast to other semiconductor materials. In this sense, diamond can serve as a model material for studying physical phenomena occurring during RD at room and elevated temperatures in other crystals. Fourth, ion implantation is still one of the most effective methods for modifying the properties of diamond. In order to be an effective semiconductor, single-crystal diamond must be precisely doped to increase conductivity without graphitizing the diamond. For such purposes, one can use ion implantation [3]. In recent years, ion implantation has been extensively used for the development of single photon emitters for purposes of quantum electronics, photonics, cryptography, magnetometry, etc. [4,5,6,7,8]. To change the conductivity and the optical properties of the surface layer of diamond, one applies ion beam technology [9].

However, defect engineering involving ion implantation is not a routine procedure, since the interaction of an ion with the diamond lattice gives rise to a number of different defects, which are distributed extremely nonuniformly over the penetration depth. Uniform damage of diamond (on thicknesses on the order of a few millimeters) is formed under the irradiation by MeV electrons. In elastic collisions, an electron most often knocks out one or two atoms; therefore, the main defects are single vacancies and interstitials. However, according to the photoluminescence (PL) spectra of electron-irradiated diamonds [10,11], under the accumulation of radiation damage, when recoiled atoms begin to interact with the defects formed earlier under irradiation, rather than with the perfect crystal lattice, a large number of various types of defects are also formed in diamond that are represented by narrow zerophonon lines in the PL spectra. Ion implantation results in a cascade of damage that generates various vacancy and interstitial complexes along the ion tracks, with the effective damaged volume of the sample being very small.

Diamonds irradiated by fast neutrons are better suited for studying the transformations of the crystal structure of diamonds under ion implantation. The interaction cross-section of neutrons with carbon atoms is small; therefore, the RD due to these interactions is uniformly distributed over the crystal volume. An atom knocked out by a neutron with energy of hundreds of keV and higher is then decelerated, knocking out new atoms, so that the main RD is formed as a result of secondary defect formation processes that result in, in addition to point defects, cascades of closely spaced displaced atoms [12]. As a result, the structure of the material is inhomogeneous at the microlevel— strongly disordered regions with a size of a few nanometers are surrounded by a weakly damaged crystalline material containing point defects. Generally, in a first approximation, all the variety of phenomena involving the effect of any ionizing radiation on diamond reduces to the case of deceleration of carbon ions. This suggests that the RD pattern of diamond exhibits a similar behavior under different types of irradiation.

Raman scattering is an informative nondestructive method for the analysis of a wide class of carbon materials. However, the application of this method for determining the composition of radiation defects and the RD level is difficult due to the limited set of experimental data and the absence of a unified approach to the interpretation of Raman an IR absorption spectra of RD diamond. Our earlier study of MeV implantation in natural diamonds [13,14] had shown that Raman scattering is a powerful technique for investigating and monitoring radiation-induced lattice modifications.

The goal of the present work is to establish, by studying Raman spectra, general regularities of the transformation of the defect structure of diamonds under their subthreshold radiation damage by fast neutrons in a wide range of fluences and during subsequent recovery annealing.

## 2. Samples and Methods

Transparent polycrystalline chemical vapor deposition (CVD) diamond films of thickness >500 μm were deposited in a MW discharge (frequency of 2.45 GHz) from a methane–hydrogen mixture in a UPSA-100 apparatus [15]. The optical and thermophysical properties of the films approached those of the best natural single crystals [16]. The N concentration in the samples was ~10^17^ cm^−3^ in the substitution position, and the concentration of hydrogen located at intercrystallite boundaries was ≤ 2 × 10^19^ cm^−3^ (determined by optical absorption in UV [17] and IR [18], respectively). CVD-diamond plates, 480 μm thick, were polished to optical quality and irradiated in a wet channel of an IVV-2M nuclear reactor, in a fast neutron flux of ~10^14^ cm^−2^ s^−1^ (with energies > 0.1 MeV) and fluences *F* = 1 × 10^18^, 3 × 10^18^, 1 × 10^19^, 2 × 10^19^ and 2 × 10^20^ cm^−2^ at 325 ± 10 K [19]. Natural type IIa and Iab diamonds were irradiated in the same reactor, with fluences of *F* = 1 × 10^20^ cm^−2^, and at 350 K in the nuclear reactor at the National Research Center “Kurchatov Institute” [20], with a fluence of *F* = 3 × 10^20^ cm^−2^, correspondingly. Under such radiation conditions, the critical dose of graphitization lies in the range of (1 ÷ 2) × 10^21^ cm^−2^. As a reference sample, we used natural diamond implanted with 335 MeV nickel ions with a fluence of *F* ≈ 5 × 10^14^ cm^−2^. In order to map the Raman spectra of Ni-implanted diamond over the entire depth of RD, a part of the sample was polished in the form of a section inclined at an angle of ≈5° [14].

Raman spectra were measured in the backscattering mode in the range of Raman frequency shifts ν_R_ = 100 ÷ 2100 cm^−1^ on a Horiba Jobin Yvon LabRAM HR spectrometer with excitation of diamond by a laser radiation (λ = 473 and 488 nm) and on a Bruker Senterra Raman spectrometer (at wavelength of λ = 532 nm). The resolution of these Raman spectrometers was Δν_R_ = 0.5–1.0 cm^−1^. The samples were annealed in an oven with graphite walls, in a vacuum of 10^−5^ Torr, at temperatures from 200 to 1700 °C (for 60 min at fixed temperature). After the heat treatment, the samples were etched in a hot H_2_SO_4_ + K_2_Cr_2_O_7_ solution at ≈ 180 °C, to remove graphite that might be formed on external surfaces.

## 3. Experiment and Discussion

### 3.1. Raman Spectra of Neutron-Irradiated Diamonds

In the unirradiated diamonds, the Raman line was at 1332.4 cm^−1^ and had a FWHM from 2 to 3.7 cm^−1^. The irradiation of two plates, cut out of the same CVD wafer, by fast neutrons with *F* = 1 × 10^18^ and 3 × 10^18^ cm^−2^ leads to a decrease in the intensity, an increase in FWHM by 4.6 and 6.3 cm^−1^, and a shift in the diamond line by 1331.2 and 1328.8 cm^−1^, respectively. In these two samples, the maximum of the band near 1630 cm^−1^, which is characteristic of the Raman spectra of radiation damaged diamonds [21,22], was at 1637 and 1634 cm^−1^, respectively. The spectra also contain a band at 1420 cm^–1^, which is attributed to intrinsic defect containing interstitial atoms [23]. The positions of the maxima and the singularities at 990, 1008, 1120, and 1235 cm^−1^ in a broad-structural low-frequency band of the Raman spectrum of unannealed diamond with *F* = 3 × 10^18^ cm^−2^ well coincides with phonon frequencies at the singular points of the Brillouin zone LA(L), LA(K) and TO(W), TO(K), LO(K), and LO(L), respectively [24,25]. In the range of 900–1300 cm^–1^, the spectrum is close to the spectrum of the PDOS of diamond (Figure 1b) [26].

A further increase in the neutron fluence (to 1 × 10^19^ cm^−2^ and higher) leads to the disappearance of the diamond peak, notwithstanding that these fluences are two orders of magnitude below the critical dose of graphitization. In part, this is explained by the fact that the Raman cross-section in diamond is much lower than that in other allotropic forms of carbon [13,27].

The dependence of the position of the maxima on fluence is fundamentally different for different bands in Figure 1a: As the fluence increases, the maximum of the low-frequency peak (the maximum at 380–400 cm^–1^) is slightly shifted to higher frequencies, whereas other singularities, conversely, are shifted by 10–30 cm^–1^ to lower frequencies and broaden. The closely spaced bands at 990 and 1008 cm^−1^ in diamonds irradiated by fast neutrons with a fluence of 1 × 10^19^ cm^−2^ and higher are not spectrally resolved. The nature of the singularity near 720 cm^−1^ remains unclear. It is possible that this singularity is attributed to the singularity in the vibrational density of states activated by the disorder of the sp^2^ phase [28]. The position, relative amplitude, and the FWHM of the band near 1620–1640 cm^−1^ are most sensitive to the value of the neutron fluence (Figure 1a). As the fluence increases to 3 × 10^20^ cm^−2^, the intensity of this band increases many times.

The shift of the maxima of the Raman bands (including the diamond band) toward the low-frequency region with increasing fluence is partially explained by the increase in the diamond lattice volume [29] due to the high concentration of defects [30]. The increase in the lattice parameter determined by the XRD method makes a contribution (of up to 6 volume %) to the swelling of RD diamond [29]. The pycnometric swelling reaches a value of 40 volume percents in diamonds in which radiation damage is below the graphitization threshold [31]. Such a strong swelling in RD diamond provides evidence in favor of the two-phase model of RD diamond. The first phase is crystalline diamond with radiation defects, and the second phase is an amorphous material with a density of about 2 g/cm^3^ [32].

Notice that all the spectra in Figure 1 are qualitatively similar irrespective of the value of fluence and the significant differences between the samples, part of which are natural diamonds and part are polycrystalline CVD diamonds. This fact is especially noticeable in samples with intermediate values of the fluence (spectra 2–5 in Figure 1a); the difference between the spectra of these samples manifests itself, as the neutron fluence increases, in the shift of the maxima and the singularities in the Raman spectrum to lower frequencies, which corresponds to an increase in the imperfection of the structure accompanied by an increase in the lattice parameter due to irradiation-induced swelling. Such behavior of the Raman spectra upon radiation damage is also characteristic of other crystals, for example, SiC [33]. Another reason for the similarity of the Raman spectra of diamonds at a high subcritical level of radiation damage is radiation-induced diffusion and partial annealing of defects [34].

According to the spectroscopic data, neutron fluencies of 1 × 10^19^ cm^−2^ and higher correspond to the range of RD levels with the dominant role of amorphization of diamond. In this range, the mechanical [32], optical [35], electrical [36], and other properties of the material exhibit a relatively small variation (saturation) with increasing fluence. Additional evidence for the subcritical amorphization of diamond is provided by the absence of not only the diamond peak but also the narrow defect-induced Raman bands in the range of 1350–1600 cm^−1^, which are attributed to the vibrations of point defects in the diamond lattice [13,14,21,23].

An RD profile in the ion-implanted samples is highly nonuniform throughout the diamond crystal. The ion-implantation process can be simulated by the Monte-Carlo software such as SRIM (Stopping and Range of Ions in Matter) code [37]. The advantages of SRIM are that it is easy to use and is widely adopted for the qualitative evaluation of ion-implanted damage degree. However, the limitations of SRIM may be significant, especially for high levels of RD. First of all, SRIM does not take into account dynamic annealing, which is the heating of the lattice occurring during implantation, due to either the implantation temperature or heating with a high dose rate. SRIM only simulates the damage of the crystalline systems, but it does not simulate the damage to an amorphous material. Here, we demonstrate that the phonon correlation length may be more suitable than SRIM vacancy concentration to characterize the defect structure in strongly radiation damaged diamonds.

By using ions with energies on the order of a hundred MeV and making confocal measurements of the Raman spectra along the beveled thin section at an angle of ≈5° prepared after implantation, it is possible to trace the transformation of the diamond crystal lattice with high spatial and spectral resolution over a wide range of radiation-induced damage with a single sample [14]. A comparison of Raman spectra for intermediate levels of RD and comparison of fast-neutron-irradiated diamonds with ion-implanted ones are illustrated in Figure 1b, which demonstrates the spectra (1) and (2) from Figure 1a and the Raman spectra of natural diamond implanted with 335 MeV nickel ions, measured confocally in the regions with a fixed level of RD [14]. As the degree of RD increases, the relative intensity of the diamond peak decreases, and the defect band with the maximum near 1630 cm^−1^ and the spectral singularity near 1000 cm^–1^ broaden and are shifted to lower frequencies. More complete data on the Raman spectra behavior in diamonds implanted with MeV ions (He, Ni, Xe, or Kr) are given in References [13,14,21], respectively. The set of Raman bands and the general shape of the Raman spectra of the samples irradiated by fast neutrons and implanted with heavy MeV ions are preserved (Figure 1 and References [13,14]). Moreover, this observation is true to both natural and CVD diamonds.

The degree of RD in diamond depends on both the irradiation temperature and the impurity concentration. For natural diamonds implanted with helium ions, such behavior was analyzed in our papers [38,39]. It is this fact that is responsible for a slight difference in the Raman spectra of natural diamonds irradiated with a fluence of 3 × 10^20^ cm^−2^ (spectra (6) and (7) in Figure 1a).

### 3.2. Phonon Confinement in Radiation-Damaged Crystals

Earlier, similar Raman spectra in RD diamonds were recorded by other scientists in CVD diamonds irradiated by fast neutrons [19,40] or implanted with high-energy ions [13,14,21,36]. In Reference [41], the authors concluded that the match between the Raman spectra of diamonds implanted with MeV helium ions and the results of DOS calculations for «amorphous diamond» [42], i.e., for clusters of nanostructured carbon with sp^3^ ordering without long-range order, provides evidence for complete amorphization of diamond under ion implantation. In Reference [21], the authors considered the possibility of interpretation of the Raman spectra of ion-implanted diamond in terms of phonon confinement [38]; however, they ruled out such a possibility on the grounds of the symmetric shape of the shifted and broadened diamond line in the Raman spectra. In the case of the limited size of the crystal, one should take into account the vibrations of the system that are related to boundary atoms. In this case, one applies the Lederman theorem, which considers the dependence of vibrational states on the size of the vibratory system and states that a macroscopic approach is applicable only in the case when the number of solutions of the vibratory system that are related to boundary atoms is greater than half of the total number of solutions [43]. Confinement manifests itself in the vibrational spectra only if the crystallite size is less than 10–15 lattice constants; therefore, from the viewpoint of phonons, a polycrystalline sample with a grain size of a few microns can be considered as a large crystal.

For the first time, a model for calculating the shape of the Raman spectrum was proposed by Richter for nanosized crystallites of silicon [44], in which, as the size of a crystallite or the phonon mean free path decreases, the main peak is shifted to lower frequencies and broadens asymmetrically. It is the discrepancy between the real shape of the diamond band and that calculated by the model of Reference [44] that served as a basis in Reference [45] for the failure to interpret the Raman spectra of ion-implanted diamond in terms of phonon confinement. However, there are a number of significant constraints on the applicability of the Richter model [44]. This model suggests that crystallites are of the same size and shape (either spherical or orthorhombic) and is absolutely inapplicable to ultrasmall crystallites due to the fact that a nanocrystal in which phonons are spatially confined by a small volume of the periodic structure is characterized by an uncertain value of the wave vector, and phonons from all points of the Brillouin zone can manifest themselves in the Raman spectrum. Moreover, the Richter model assumes that the wave functions are isotropic in all crystallographic directions (spherical model). This approach is inapplicable to diamond, in which the dispersion of phonon branches is not only changed in magnitude but also can become negative [46]. According to the model calculations of Raman spectra for nanodiamonds [45], with regard to all three optical modes, a shift of the diamond peak in the absence of mechanical stresses should not be observed until the sample size (phonon coherence length) decreases down to 5 nm. The maximum value of the reciprocal vector *k* that takes part in the formation of Raman spectra is on the order of 1/*L*, where *L* is the crystallite size or the mean free path of phonons. Confining the phonon to a smaller volume in real space leads to a larger uncertainty in its momentum. In diamond, phonons from the entire Brillouin zone contribute to the Raman signal for *L* = 2 nm or less [45]. Based on these relations, in Reference [45], the authors explained the broad band with a maximum near 1250 cm^–1^ in the Raman spectra of nanodiamonds by the phonon confinement.

In an amorphous material, there is no long-range order; therefore, the size *L* is on the order of the interatomic distance. Under amorphization, the degree of disorder is so large that all vibration modes can take part in the first-order Raman spectrum. Such behavior of the Raman spectra of crystallites under their RD has been described in the publications on electron-irradiated NaCl crystals [47], ion-implanted nanocrystals of Ge [48] and GaAs single crystals implanted with light ions [49], single crystal Si implanted with Au ions at low temperature [50], MoS_2_ monolayers implanted with Mn^+^ ions [51], and in neutron-irradiated 6H–SiC crystals [52]. Note that not all radiation-damaged materials [47,48,49,50,51,52] in which phonon confinement was observed experimentally were amorphous. The shift and the broadening of high-frequency lines of optical vibrations were observed not only in nanocrystalline and amorphous materials, but also in strongly defective materials. In this case, the phonon mean free path *L*_ph_ is determined not by the size of nanocrystallites but by the average distance between defects and the nonuniformity of the distribution of atoms in disordered semiconductor solid solutions and alloys (*L*_ph_ represents the coherence length and is therefore a measure of the distance between dislocations, twin boundaries, stacking faults, vacancies, interstitials, impurities, and other defects within the crystal lattice). In an RD region, when *L*_ph_ decreases to a few interatomic distances, the selection rules are relaxed. Along with the shift and the broadening of the main band, this leads to the manifestation of the whole density of phonon states in the Raman spectra. We believe that precisely such a picture is observed in the Raman spectra of natural and CVD diamonds irradiated by fast neutrons ([13] and Figure 1) or implanted with high-energy ions [14] (Figure 1).

### 3.3. Raman Spectra of Amorphous Diamond

In diamond, the phonon frequencies at singular points range from 1332.5 cm^−1^ (LO, TO (Г)) to 553 cm^−1^ (TA(L)), which cannot explain the presence of the lowest frequency bands in the Raman spectrum (Figure 1). The broad low-frequency band with a maximum at about 400 cm^−1^ (Figure 1) coincides with the band observed in natural diamonds implanted with helium ions [21,39], as well as in nanosized diamonds [53]. Historically, such a peak in disordered materials has been called the boson peak. In Si, the boson peak coincides with the transverse acoustic (TA) mode; therefore, the density of vibrations near the boson peak is simply the density of TA vibrations [54]. Note that there is no consensus on the nature of the boson peak in the literature [55,56,57]. After the authors of Reference [58] concluded, on the basis of the measurements of the properties of glasses, that the boson peak is equivalent to van Hove singularities of longitudinal acoustic vibrations (rather than quasilocal vibrations of defects), the discussion on the nature of the boson peak seemed to be completed [59]. However, later, the same authors [60] expressed doubts about their own conclusions made in their earlier work [58]. Moreover, they tried to verify these conclusions in the measurements of diamond amorphized by neutron irradiation, in which a van Hove singularity is located near 75 meV (600 cm^−1^). However, after exceeding the critical dose of graphitization as a result of irradiation, they obtained amorphous carbon with a high fraction of the sp^2^-component [60]. This is an important result because, according to our investigations [39], amorphous diamond is unstable under normal conditions and can exist only in the form of nanoinclusions due to the pressure from the diamond matrix.

Today, calculations of the density of phonon states for amorphous sp^3^-carbon (amorphous diamond) are missing in the literature. In Reference [61], the authors calculated the density of phonon states for tetrahedral amorphous carbon, whose spectral shape is noticeably different from that observed in the Raman spectra of RD diamond. The point is that the authors of Reference [61] considered amorphous diamond as a material with predominantly sp^3^-bonds, as well as a material with a high (up to 10–15%) component of sp^2^-bonds, which is responsible for both the high-frequency boundary of the calculated spectrum and a significant contribution of the sp^2^-component to the PDOS. According to the Raman spectra (Figure 1), the character of variation of the Raman spectra under annealing (see below), the mechanical properties [39], and neutron diffractometry data [19] in diamonds irradiated with a fluence below the critical dose of graphitization, the fraction of sp^2^-carbon is negligible, and the calculations in Reference [61] are unsuitable for the analysis of Raman spectra for a subcritical level of RD.

For disordered and amorphous Ge, the calculations of PDOS were performed in Reference [62], including isolated germanium nanoparticles, nanocrystals, and nanoparticles embedded in glasses, as well as nanoglasses, which represent a class of materials synthesized by consolidation of glassy nanoparticles [63]. Germanium was chosen as a prototype covalent material with crystalline and amorphous modifications. The authors of Reference [62] specially mentioned that the results of their calculations can be extrapolated to other crystals, including crystals with the lattice closest to that of diamond. In Reference [62], the authors demonstrated that the vibrational modes are sensitive to the changes in the nanostructure, compared these modes with the vibrational modes in crystalline and amorphous bulk materials, and established how the PDOS changes depending on the size of nanoparticles due to the surface tension and confinement phenomena. In nanocrystalline objects, which can be represented as an ensemble of nanoparticles connected by «glassy» regions of grain boundaries, as the particles size decreases, the maximum of the low-frequency region of the PDOS is shifted to lower frequencies, and the bands themselves become broader. Simultaneously, the grain boundaries provide the most significant contribution to the increase in scattering in the region of acoustic modes of the PDOS, due to the general broadening of the PDOS in these regions. A grain size below 1.1 nm leads to grain instability in Ge and, as a result, to an amorphous structure. This value is in good agreement with the minimum grain size of Ge (1–2 nm) observed in the experiments of Reference [64]. The PDOS of such an amorphous structure differs from that calculated for bulk glass: The acoustic branch is shifted to the low-frequency region and broadens, a result that was attributed in Reference [64] to the excess free volume of Ge in the amorphous state.

For diamond, a qualitatively similar result was obtained [65]. The analysis of the behavior of the broad low-frequency band in the Raman spectra of radiation-damaged diamond (depending on the damage level of diamond) suggests that, just as in Ge, this band is attributed to amorphized diamond. This fact is evidenced by the high-frequency shift of this band with increasing fluence and its behavior under annealing (see below). According to Reference [45], the estimated phonon mean free path *L*_ph_ is on the order of 1 nm, or three lattice parameters, which does not contradict the calculations performed in Reference [65]. A similar result was obtained by the authors of Reference [54], who investigated amorphous porous silicon; they showed that the coefficient of interaction between light and acoustic vibrations (in contrast to the coefficient of interaction with optical vibrations) contains an additional factor—the square of the reciprocal correlation length of vibrational excitations; in other words, the intensity of light scattering by acoustic phonons has an additional dependence on the degree of disorder. Due to this effect, the boson peak located in the acoustic range of the Raman spectrum of amorphous porous silicon is more sensitive to the structural order than the optical mode. According to the Raman spectroscopy data, the conclusions of Reference [54] for amorphous porous silicon are also valid for RD diamond, the low-frequency band in the Raman spectra of unannealed samples of which corresponds to vibrations of amorphous diamond or, in other words, of diamond with a phonon mean free path on the order of 1 nm, which is confirmed by the behavior of the Raman spectra under thermal annealing.

### 3.4. Annealing Behavior of Fast Neutron-Irradiated Diamonds

The heating of irradiated diamonds removes damage by adding energy to the system, so that, due to the increased mobility of defects, the carbon atoms find lower-energy sites to occupy in the lattice, thereby restoring the diamond crystal structure. Figure 2 demonstrates the transformations, under annealing, of the Raman spectra of CVD diamond irradiated by neutrons with a fluence of 2 × 10^19^ cm^−2^. As a result of annealing at 550 °C, the diamond peak, which is missing in unannealed samples, clearly manifests itself in the spectra (Figure 2), its intensity increases with annealing temperature, and its FWHM decreases. As the annealing temperature increases, the maximum of this peak is shifted from 1285 cm^−1^ (annealing at 450 °C) to 1317 cm^−1^ (annealing at 900 °C), and then to 1328.5 cm^−1^ (annealing at 1580 °C, Figure 2a).

The effect of annealing at temperatures below 700 °C is more clearly demonstrated in the difference Raman spectra measured under the same conditions (Figure 2b). As the annealing temperature increases, the intensity of the broadened diamond band and the structural band with a maximum at 1630 cm^−1^ [16,23] shifted to lower frequencies significantly increases. This points to a decrease in the degree of disorder in the material, the relaxation of elastic stresses in it, and a decrease in the intensity of the boson peak in the Raman spectra with a maximum near 400 cm^−1^. Simultaneously, the amplitude of the Raman signal in the range 900–1100 cm^−1^ decreases, which, according to the calculations in References [62,65] of the PDOS for amorphous materials with sp^3^ bonds, points to the increase in *L*_ph_. In the difference spectra (Figure 2b), additional singularities at frequencies of about 1088 and 1126 cm^–1^ become visible. The positions of these singularities coincide with high accuracy with the frequencies of TO-phonons at the points X and K, respectively. The appearance of these singularities in the Raman spectra is also attributed to phonon confinement due to a decrease in the phonon mean free path *L*_ph_ in RD diamond. The same two singularities are also observed in the Raman spectra after annealing at 800–1000 °C (Figure 2a), which leads to a decrease in the level of RD in diamond.

As the diamond peak is recovered, high-frequency bands arise in the Raman spectra that are attributed to the vibrations of defects in the diamond crystal lattice. Calculations in Reference [66] show that the defects with frequencies >1400 cm^−1^ in the spectrum of diamond are highly localized, while the narrow half-width of the peaks indicates that they are caused by point defects, rather than by extended defects. In the high-frequency region, rather narrow bands with maxima at 1430 and 1480 cm^−1^ appear and increase in intensity, which were almost indistinguishable in the original sample after neutron irradiation with a fluence of 2 × 10^19^ cm^−2^. As the annealing temperature increases, these bands are shifted toward their positions in weakly damaged diamond, i.e., to 1450 and 1501 cm^−1^, as the swelling of the samples decreases [14]. As the annealing temperature increases, the Raman band near 1610 cm^−1^ is shifted to 1630 cm^−1^, and its shape and intensity change, with the intensity attaining its minimum at 625 °C. For a further increase in temperature, this band splits into several components and is completely annealed at temperatures much higher than 1000 °C, as described in Reference [10]. Starting from the annealing temperature 450 °C, there appears and grows a relatively narrow (FWHM~25 cm^−1^) line near 1800 cm^−1^ (Figure 2), which is the highest-frequency line in this spectrum; in ion-implanted diamonds, this line shifts to 1814 cm^–1^ as the RD level decreases [14]. All four abovementioned bands are characteristic of radiation-damaged diamonds. In particular, they are observed in diamonds implanted with heavy MeV ions [10], where their intensities relative to the diamond band have a threshold character, depending on the calculated vacancy concentration. More precisely, these sharp bands begin to appear in the Raman spectra of ion-implanted diamonds only for a calculated vacancy concentration of ≈1.5 ×·10^20^ cm^−3^ [14].

Different tentative attributions have been formulated for the 1430 (1450) and 1480 (1501) cm^−1^ peaks [14], involving vacancy [41,65], intrinsic/nitrogen interstitial defects [67,68], or a vacancy or a divacancy surrounded by conjugated single and double carbon–carbon bonds (the R4/W6 center) [21,69]. Recent calculations [70] based on the B3LYP hybrid implementation of density functional theory [71] show the absence of peaks in the region of 1490 cm^−1^ in the spectra of both divacancy V_2_ and V—C=C—V. According to Reference [72], the calculated Raman-active high-frequency local vibrational modes of di-interstitials (1461, 1495, 1813, and 1826 cm^−1^) agree well with the bands often observed in the Raman spectra of radiation-damaged diamonds. The interpretation of Reference [72] is evidenced by the fact that the bands 1450, 1501, and 1814 cm^−1^ exhibit identical temperature dependence under annealing of diamonds irradiated by fast neutrons (Figure 2a). Experiments on establishing the nature of narrow defect-induced bands are in progress and will be reported in forthcoming publications. The low-frequency boson peak at 400 cm^−1^, which is characteristic of the amorphous diamond phase, decreases during annealing and becomes hardly visible at annealing temperatures of 800–1000 °C (Figure 2a). All of this is explained by the slow recovery of the diamond crystalline phase, an increase in its fraction due to the recrystallization of amorphous domains, and a decrease in the concentration of point defects and mechanical stresses in this phase.

The low-frequency boson peak at 400 cm^−1^, which is characteristic of the amorphous diamond phase, decreases during annealing and becomes hardly visible at annealing temperatures of 800–1000 °C (Figure 2a and Figure 3a). All of this is explained by the slow recovery of the diamond crystalline phase, an increase in its fraction due to the recrystallization of amorphous domains, and a decrease in the concentration of point defects and mechanical stresses in this phase.

### 3.5. Evidence of New Raman Bands after High-Temperature Annealing of RD Diamonds

Along with the decrease of the boson peak, bands with maxima at 257 (260) and 460 cm^−1^ appear in the Raman spectra starting from the annealing temperature of 550 °C, which, together with the band at 725 cm^−1^, become dominant in the low-frequency part of the Raman spectra after annealing at 900–1000 °C (Figure 2a). The intensity of these three bands increases faster than the intensities of both the boson peak and the recovering diamond peak and reaches a maximum after annealing in the temperature range of 1000–1200 °C. The shape of the Raman spectra of the diamonds irradiated with neutron fluences from 1 × 10^19^ to 3 × 10^20^ cm^−2^ is almost independent of the excitation wavelength, the impurity composition of diamonds, and their origin (CVD or natural) (Figure 3).

Earlier, Raman spectra of similar shape (Figure 3b) were observed in diamonds implanted with helium ions with energy 2 MeV [21] and in diamonds irradiated by fast neutrons with a fluence of (1.3–2.7) × 10^21^ cm^−2^ at temperature 450–530 °C [73]. In Reference [21], broad peaks at 250 and 480 cm^−1^ appear in the Raman spectra after annealing at about 825 °C, have their highest intensity at 945 °C, and decrease again at 1075 °C, while the peak at 750 cm^–1^ has its maximum intensity at about 600 °C. In Reference [21], the annealing at 1150 °C led to the disappearance of all three additional bands in the Raman spectra, although the intensity of the diamond band remained low. The system of bands with maxima at 252, 490, 650–750, and 1470 cm^−1^ in Reference [73] also disappeared in the Raman spectrum after annealing at 1150 °C. In Reference [73], the authors suggested that this system of bands in the Raman spectra is attributed to the vibrations of the intermediate carbon phase, which consists of both sp^3^- and sp^2^-bonded carbon atoms (the ratio of the sp^3^ and sp^2^ bonds is 1/2) and has a deformed hexagonal lattice with corrugated graphite-like layers crosslinked by strong sp^3^ bonds. In References [73,74], the authors calculated vibrational spectra for the intermediate carbon phase, according to which 24 optical frequencies out of 45 correspond to Raman-active modes. The Raman-active modes can be divided into groups of closely spaced frequencies near 250 cm^−1^ (three modes), 500 cm^−1^ (three modes), 650–850 cm^−1^ (five modes), 1050–1390 cm^−1^ (10 modes), and 1414–1530 cm^−1^ (three modes).

The measurements of the Raman spectra of diamonds irradiated by fast neutrons with fluences from 1 × 10^19^ to 3 × 10^20^ cm^−2^ (Figure 2, Figure 3 and Figure 4) do not confirm the calculated data from References [73,74]. The intensity of the group of bands 260, 495, and 730 cm^−1^ significantly depends on the fluence of neutrons and the annealing temperature, both with respect to the signal in the range 1050–1390 cm^−1^ and to the three most intense bands in the high-frequency region (1450, 1470, and 1500 cm^−1^); this fact does not allow one to attribute these bands to the vibrations of the same defects or carbon phases. The bands at 260, 495, and 730 cm^−1^ have no fine structure, which also disagrees with the data of [73,74]. At the same time, in the Raman spectra of diamonds irradiated by fast neutrons and annealed at temperatures above 1000 °C, the singularity near 1070 cm^−1^ is spectrally resolved, and its frequency coincides with the TO-phonon frequency at the X point of the Brillouin zone. The appearance of this singularity in the Raman spectrum is attributed to the high concentration of defects and, hence, to the phonon confinement effect in RD diamond. It is worth noting that the bands at 260, 495, and 730 cm^−1^ are missing in the Raman spectra of unannealed diamonds implanted with heavy MeV ions, and the appearance of these bands in the spectra requires high-temperature annealing.

In contrast to the samples investigated in References [21,73], annealing at temperature 1150 °C did not lead to the disappearance of the bands at 260, 495, and 730 cm^−1^ in the Raman spectra (Figure 2, Figure 3 and Figure 4); this can be attributed to the lower level of radiation damage of the diamonds in References [21,73]. Moreover, annealing at higher temperatures has shown that the centers responsible for the system of bands 260, 495, and 730 cm^−1^ persist after annealing at least 1550 °C (Figure 2a and Figure 4).

An increase in temperature to about 1300 °C gives rise to another system of bands in the Raman spectra that have similar temperature dependence of intensity: 230, 500, 530, 685, and 760 cm^−1^. These bands are indicated by arrows in spectrum (6) in Figure 4. The spectral position of the maxima of this system of five bands is hardly changed under increasing annealing temperature, and the intensity of the bands with respect to the diamond line reaches a maximum at 1000–1100 °C (Figure 2a and Figure 4). Further annealing leads to the recovery of the diamond line, and the system of bands persists in the Raman spectra up to the maximum admissible annealing temperatures of diamond under normal conditions (1700 °C [75]); however, the intensity of this line with respect to the diamond peak decreases by about a factor of 10 in this case. A different temperature dependence is observed in another group of bands with maxima at 335, 1390, 1415, and 1740 cm^−1^. The intensity of all of these four bands synchronously increases in the temperature range of 1350–1650 °C, much faster than the intensity of the diamond peak and simultaneously with the strengthening of the 580 nm band of H19 center in PL spectra [76].

The nature of the additional bands in the Raman spectra of diamonds irradiated by fast neutrons and annealed at 1300–1700 °C remains unclear. Judging by the behavior of the Raman spectra as a function of the annealing temperature, here we deal with a Raman signal of rather stable shape that consists of a few rather narrow bands, which are indicated by arrows in Figure 4. The shape of the Raman spectrum of RD diamond after high-temperature annealing is the same as that of CVD and natural diamonds and is almost independent of their impurity composition and, in a certain range of fluencies, of the initial level of RD. In spite of the high level of RD and high temperature of vacuum annealing, no signs of graphitization of samples were detected—the Raman spectra contain an intense diamond peak, while the D- and G-bands, which are characteristic of sp^2^-carbon, are completely missing. The presence of bands with frequencies above 1400 cm^–1^ in the Raman spectra (Figure 2a and Figure 4) indicates the presence of C=C groups in the structure of inclusions formed in the bulk of diamonds irradiated by fast neutrons with high subcritical fluences and annealed at high temperatures. This shape of spectra (Figure 4) is uncharacteristic for all known pure carbon materials [77].

Transmission electron microcopy is an excellent method for direct examination of the radiation-induced defect formation and the evolution of diamond crystal structure as a result of fast neutron irradiation or ion implantation followed by annealing. The phase transformation from ion-implanted crystalline diamond to amorphous carbon was considered in References [78,79,80,81]. These investigations mainly focus on the process of graphitization of diamond at RD level above the critical value when the diamond lattice is too damaged to be repaired [82]. The TEM observation in Reference [80] suggests that the damaged layer in C-implanted diamond does not completely recrystallize back to diamond by solid phase epitaxial regrowth from the underlying crystalline substrate.

At the same time, there are no extended defects detected by TEM in the bulk of diamond with RD level below the critical doze [81]. According to Reference [81], the most likely explanation of the fact that extended defects (such as the {311} defect and dislocation loops typical of silicon and germanium lattices [83,84,85]) are not forming in the diamond lattice is that the strong covalent bond between the carbon atoms may make formation of extended defect configurations thermodynamically unstable.

Meanwhile, the numerous extended planar defects up to 5 nm in length were observed in the HRTEM images in Reference [80], just below the interface of the implanted region and more toward the diamond substrate at the end of the range of the carbon ions. Measurements of the Raman spectra were not performed in Reference [80]. We believe that the additional bands in the Raman spectra of diamonds irradiated by fast neutrons with high subcritical fluences can be attributed to extended defects, including defects similar to those observed in ion-implanted diamonds [80].

## 4. Conclusions

Using the Raman spectroscopy data for CVD and natural diamonds irradiated by fast neutrons with fluences below the critical level of graphitization and annealed at temperatures of up to ~1650 °C, we have investigated the processes of nanostructuring and defect formation in a diamond lattice. We have shown that the Raman spectra of fast-neutron-irradiated and ion-implanted CVD and natural diamonds contain the same set of bands, which provides evidence for the generality of RD processes in diamond. We have found the following:(i)The shape of the Raman spectrum of diamonds irradiated by fast neutrons with a fluence of 3 × 10^18^ cm^−2^ or higher is determined by the phonon confinement effect due to the high concentration of intrinsic radiation defects. A measure of radiation damage in such samples is given by the phonon coherence length, L_ph_. A decrease in L_ph_ to ~2 nm leads to the removal of the restriction that only phonons from the center of the Brillouin zone can contribute to the Raman spectra and gives rise to a band with a shape close to that of the PDOS of diamond in the Raman spectrum.(ii)The broad low-frequency band with a maximum near 400 cm^−1^ (the boson peak) in the Raman spectra of RD diamonds points to the formation of an amorphous sp^3^-phase with L_ph_~1 nm in RD diamonds. The boson peak persists in the Raman spectra of RD diamond up to annealing temperatures of 800–1000 °C, i.e., up to about the same temperatures at which di-interstitials (bands 1450, 1501, and 1814 cm^−1^) and the intrinsic defects responsible for the structural band near 1640 cm^−1^ are annealed. As the annealing temperature increases, the corresponding singularities in the Raman spectra are monotonically shifted to higher frequencies and approach the position of the bands in the PDOS spectra of diamond that correspond to the frequencies of optical and acoustic phonons at the singular points of the Brillouin zone.(iii)The fact that the Raman spectra of RD diamonds contain both the diamond and boson peaks confirms the hypothesis on the two-phase microstructure of RD diamond, in which nanosized amorphous regions are separated by crystalline regions with radiation defects and stabilized by the internal pressure.(iv)Annealing at high temperatures does not lead to the complete recrystallization of the diamond lattice. In the crystalline diamond phase, radiation defects represented in the Raman spectra by three groups of bands (first group = 260, 495, and 730 cm^−1^; second group = 230, 500, 530, 685, and 760 cm^−1^; and third group = 335, 1390, 1415, and 1740 cm^−1^), whose relative intensities synchronously change under annealing, experience restructuring during annealing. The maximum frequencies and FWHM of these bands are almost insensitive to their intensity, degree of RD, and the defect–impurity composition of the original diamond, and the bands themselves are attributable to vibrations of extended defects, which are stable in RD diamonds at least up to a temperature of 1700 °C.

## Figures and Tables

**Figure 1 nanomaterials-10-01166-f001:**
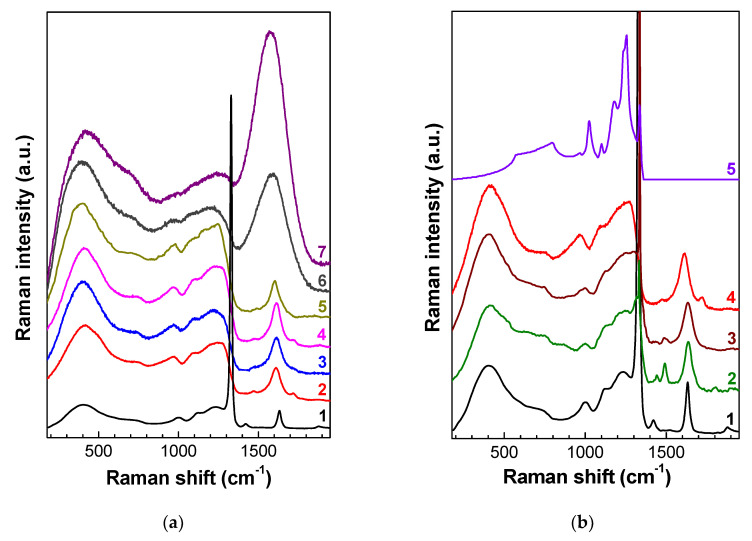
(**a**) Raman spectra of chemical vapor deposition (CVD) diamonds irradiated by fast neutrons: fluences of 3 × 10^18^ cm^−2^ (1), 1 × 10^19^ cm^−2^ (2), 2 × 10^19^ cm^−2^ (3), and 2 × 10^20^ cm^−2^ (5). Raman spectrum of natural diamonds with a neutron fluence of 10^20^ cm^–2^ (4) and 3 × 10^20^ cm^–2^ (6 and 7). Spectrum (1) is recorded with excitation at 532 nm; (3) and (5), with excitation at 488 nm; and other spectra, with excitation at 473 nm. The spectra are vertically shifted for clarity. (**b**) Raman spectra of CVD diamonds irradiated by neutrons 3 × 10^18^ cm^−2^ (**1**) and 1 × 10^19^ cm^−2^ (4) and of natural diamond implanted with 335 MeV nickel ions in the region with calculated vacancy concentrations of 4.3 × 10^21^ cm^−3^ (2) and 1.1 × 10^22^ cm^−3^ (3) [14]. (5) The density of phonon states in diamond [19].

**Figure 2 nanomaterials-10-01166-f002:**
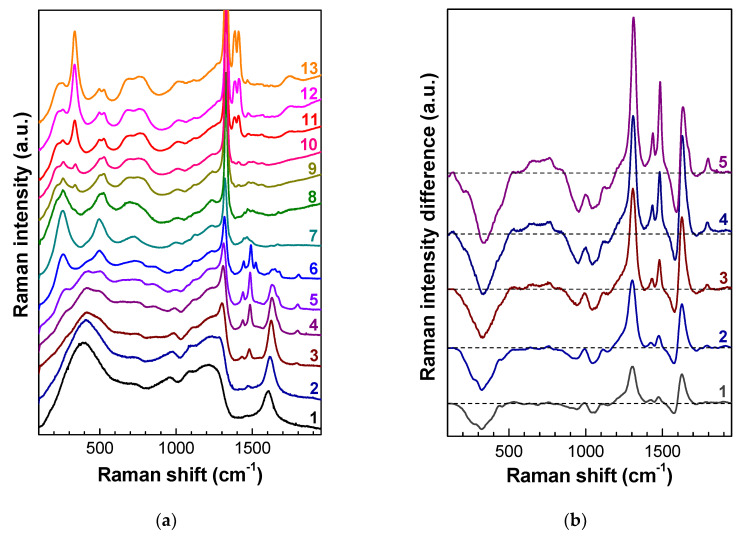
(**a**) Transformation of the Raman spectra of neutron-irradiated diamond with a fluence of 2 × 10^19^ cm^−2^, depending on the annealing temperature. The annealing temperatures for the spectra were (1) 100 °C, (2) 450 °C, (3) 550 °C, (4) 700 °C, (5) 800 °C, (6) 900 °C, (7) 1005 °C, (8) 1285 °C, (9) 1375 °C, (10) 1465 °C, (11) 1520 °C, (12) 1555 °C, and (**13**) 1580 °C. The spectra were recorded on the growth side of CVD diamond with excitation at 488 nm and vertically shifted for clarity. (**b**) Difference spectra between spectrum (1) in (a) and the spectra of the same sample after annealing at (1) 400 °C, (2) 450 °C, (3) 550 °C, (4) 625 °C, and (5) 700 °C.

**Figure 3 nanomaterials-10-01166-f003:**
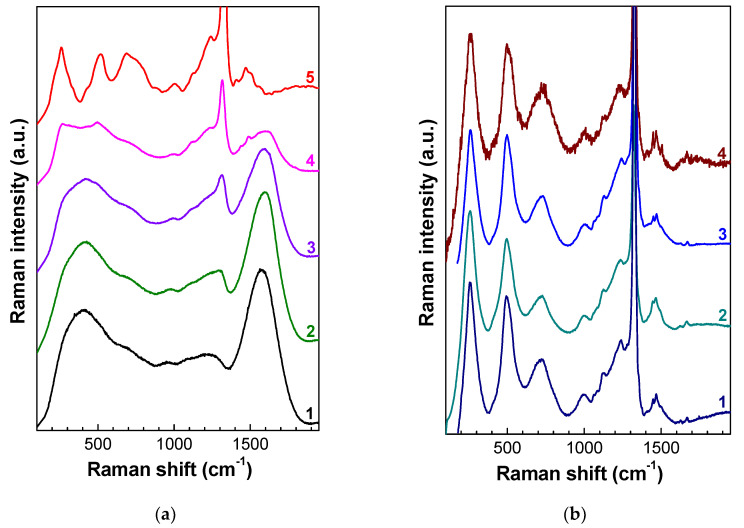
(**a**) Transformation of the Raman spectra of natural diamond irradiated with fast neutrons with fluence of 3 × 10^20^ cm^-2^ and annealed at (1) 100 °C, (2) 600 °C, (3) 810 °C, (4) 910 °C, and (5) 1310 °C. (**b**) Raman spectra of neutron-irradiated diamond and annealed at T_ann_ diamonds: (1) *F* = 2 × 10^19^ cm^−2^, T_ann_=1005 °C; (2) *F* = 1 × 10^20^ cm^−2^, T_ann_ = 1125 °C; (3) *F* = 1×10^19^ cm^−2^, T_ann_ = 1175 °C; and (4) *F* = 2 × 10^20^ cm^−2^, T_ann_ = 1150 °C. The spectra are vertically shifted for clarity.

**Figure 4 nanomaterials-10-01166-f004:**
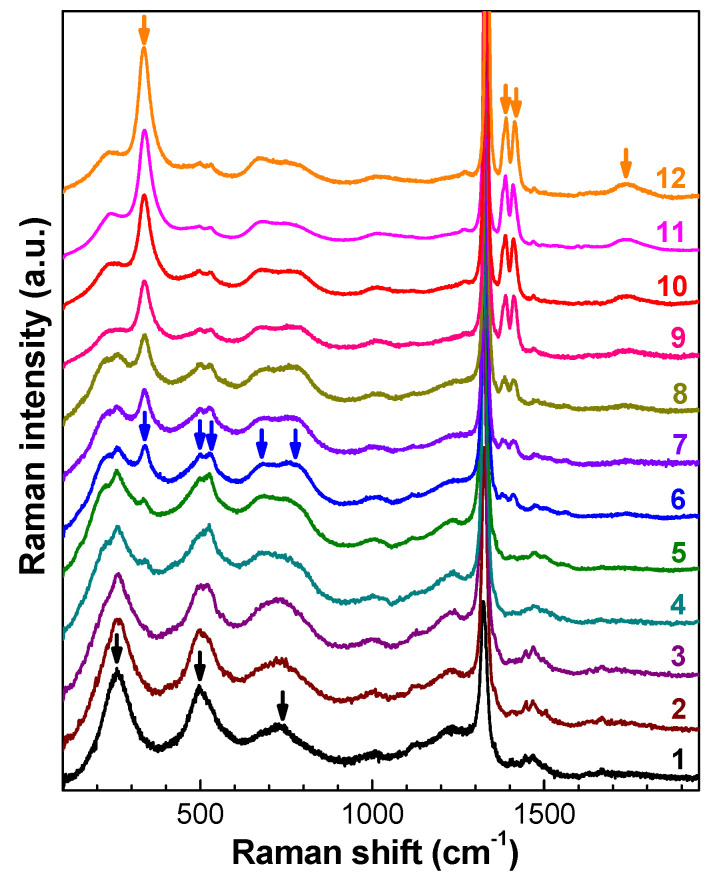
Transformation of the Raman spectra of neutron-irradiated diamond with a fluence of 2 × 10^20^ cm^−2^, depending on the annealing temperature. The annealing temperatures for the spectra were (1) 1080 °C, (2) 1150 °C, (3) 1250 °C, (4) 1300 °C, (5) 1375 °C, (6) 1465 °C, (7) 1505 °C, (8) 1535 °C, (9) 1610 °C, (10) 1650 °C, (11) 1665 °C, and (12) 1680 °C. The spectra were vertically shifted for clarity. The arrows indicate three groups of bands whose relative intensities synchronously change under annealing.

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
