# Peer review of "Probing the Nanostructure of Neutron-Irradiated Diamond Using Raman Spectroscopy"

_nanomaterials, 2020, doi:10.3390/nano10061166_

Round 1

Reviewer 1 Report

The manuscript entitled "Probing the nanostructure of neutron-irradiated diamond using Raman spectroscopy" brings new information on the behavior of the defects in the diamond during neutron irradiation and annealing. There are a couple of issues mentioned below, but generally, I found the paper interesting and worth publishing.

The text lacks structuring. Subsections within the experiment and discussion section would be very helpful. Some paragraphs are very long. For example, the first paragraph in the introduction has 42 lines, or conclusions are just one paragraph. This makes the paper difficult to read. A table summarizing all mentioned Raman bands and their most probable assignments would also be helpful.

Although the discussion is filled with relevant references, there are only 10 in the introduction. There are several claims that could use support in a reference. (e.g. the ion implantation doping of diamond)

How was the hydrogen concentration in the samples evaluated?

Why do you use both natural and CVD diamonds? There is no comparison of their behavior in the conclusion.

Why did you use a different excitation wavelength for some samples? This makes comparing their spectra difficult (especially the PL band, which I actually didn't recognize in your presented spectra, although it is discussed).

Why were some samples polished to form a section inclined 5o? There are no depth-related results.

Author Response

Dear colleague.

Thank you very much for your review. Your questions and comments turned out to be fair and allowed us to improve the text of the article. Herein we try to give comments and answers on your review.

Point 1: The text lacks structuring. Subsections within the experiment and discussion section would be very helpful.

Response 1: We add four subsections in the Experiment and discussion section.

Point 2: Some paragraphs are very long. For example, the first paragraph in the introduction has 42 lines, or conclusions are just one paragraph. This makes the paper difficult to read.

Response 2: The Introduction and Conclusions texts are fragmented and paragraphed.

Point 3: A table summarizing all mentioned Raman bands and their most probable assignments would also be helpful.

Response 3: There are certain difficulties with compiling such a table. The fact is that the positions of the band maxima depend both on the neutron fluence and on the annealing temperature. Therefore, these tables should be supplemented with appropriate graphical dependencies, which overloads the article material with unnecessary details. In addition, the presence of defective bands in the spectrum is determined by the processes of their generation and annihilation. We will take into account your wish in the next article, in which the main subject of research will be defects in radiation-damaged diamond, which manifest themselves in narrow bands in the IR absorption and Raman spectra.

Point 4: Although the discussion is filled with relevant references, there are only 10 in the introduction. There are several claims that could use support in a reference. (e.g. the ion implantation doping of diamond).

Response 4: We have supplemented the references in the Introduction section with four recently published articles.

Point 5: How was the hydrogen concentration in the samples evaluated?

Response 5: We use IR spectra to determine bonded hydrogen content in CVD diamond films (reference [18]).

Point 6: Why do you use both natural and CVD diamonds? There is no comparison of their behavior in the conclusion.

Response 6: Our studies have shown that there is no significant difference in the Raman spectra of pure natural and CVD diamonds. The intercrystalline boundaries in CVD diamonds (with diamond crystallite sizes of the order of several tens of micrometers) do not significantly affect the diffusion and annihilation of radiation defects. The main factors affecting the RD level of diamond is the concentration of nitrogen impurity (at the level of 1019 cm-3 and higher), which actively interacts with vacancies and other intrinsic defects. We refer this fact in the discussion of the Raman spectra of natural diamond microcrystals irradiated with fast neutrons with a fluence of 3*1020 cm-2 and in our paper [39].

Point 7: Why did you use a different excitation wavelength for some samples? This makes comparing their spectra difficult (especially the PL band, which I actually didn't recognize in your presented spectra, although it is discussed).

Response 7: Irradiation of diamond by fast neutrons can lead to a substantial increase in the photoluminescence bands with zero-phonon lines, in particular, at 470, 484, 491, 503, 504, 524, 535, 555, 575, 580, 590, 594, 638 nm, etc., some of which had intense phonon wings. The relative intensities of the PL bands of these centers are determined by the impurity content of the irradiated diamonds, the degree of RD, the temperature of the subsequent annealing, and the excitation wavelength. By choosing the excitation wavelength, we minimized the manifestation the effect of intense photoluminescence in the Raman spectra. At the same time, the Raman spectra in RD diamonds were very insignificantly dependent on the wavelength of the exciting radiation.

Point 8: Why were some samples polished to form a section inclined 5°? There are no depth-related results.

Response 8: We used a natural diamond implanted with 335 MeV nickel ions as a reference sample. The region of maximum radiation damage for nickel ions with such energy is at a depth of about 30 micrometers. The presence of oblique thin section at a small angle allows us to study the radiation damage region of diamond over its entire depth in details. We presented two Raman spectra of this sample in order to show that the shape of the spectrum of RD diamonds does not depend on the method of radiation damage, and also to demonstrate that a significant change in the shape of the Raman spectrum when moving from a sample with a neutron fluence of 3*1018 to 1*1019 cm-2 occurs rather monotonously.

Please see modified version of the article in the attachment.

Reviewer 2 Report

The publication is clear and clearly argued.
The conclusions are correct and publication is worthy of Nanomaterials.
The complete references correspond well to the study. It would have been interesting to make a table of the different attributions of the Raman modes.

Author Response

Dear colleague.

Thank you very much for your review!

Reviewer 3 Report

The work contains an excellent introduction to the subject. The content of the experimental results is well supported by references to theoretical studies. Work very widely described and interpreted. The precise interpretation of Raman spectra subjected to both irradiation and the heating process. The results make a significant contribution to understanding induced defects and changing the properties of diamond structures for use in microelectronics.

Author Response

(The authors gave the same response as above.)

Reviewer 4 Report

The article titled "Probing the nanostructure of neutron-irradiated diamond using Raman spectroscopy" written by Andrey A. Khomich, Roman A. Khmelnitsky, Alexander V. Khomich describes a neutron-irradiated diamond and deep analysis using Raman spectroscopy. The results seem interesting and a deep Raman analysis qualifies the article for acceptance in its current form.

Author Response

(The authors gave the same response as above.)
